# Communicating Risks and Food Procedures through a Visual Poster for Caregivers of Patients with Dysphagia in Inpatient Care: Usability and Impact

**DOI:** 10.3390/healthcare12020148

**Published:** 2024-01-09

**Authors:** Rafaela Nogueira Neves, Maria Assunção Matos, Irene P. Carvalho

**Affiliations:** 1Medical Psychology Unit, Department of Clinical Neurosciences and Mental Health, Faculty of Medicine, University of Porto, Alameda Prof. Hernâni Monteiro, 4200-319 Porto, Portugal; 2Department of Speech and Language Therapy, Aveiro School of Health Sciences, University of Aveiro, Campus Universitário de Santiago, Agras do Crasto, Edifício 30, 3810-193 Aveiro, Portugal; maria.matos@ua.pt; 3CINTESIS@RISE, Aveiro School of Health Sciences, University of Aveiro, 3810-193 Aveiro, Portugal; 4CINTESIS@RISE, Faculty of Medicine, University of Porto, 4200-450 Porto, Portugal

**Keywords:** dysphagia, communication signs, inpatient care, caregivers, health

## Abstract

Food-related procedures are a part of rehabilitation interventions for dysphagia. However, studies show that professional-caregiver communication is often lacking in dysphagia, risking caregivers’ knowledge, understanding, and practice of those procedures, with negative consequences for patient safety and rehabilitation. The aim of this study was to evaluate caregivers’ perspectives about the utility of a poster designed to communicate dysphagia-related risks and food procedures for caregivers of patients in inpatient care. The impact of caregivers’ exposure to the poster on patients’ dysphagia-related health was additionally explored. The poster was placed by the beds of a randomly assigned group of patients (*n* = 21). Their caregivers responded to a questionnaire about the poster’s utility. In addition, to explore whether the caregiver exposure to the poster could already have some effect on patient dysphagia-related health, patient risk of aspiration, food swallowing capacity, nutritional status, and oral cavity health were assessed before and one month after placement of the poster, and the poster-exposed group was compared with a (randomly-assigned) non-exposed group (*n* = 21). Data were analyzed with descriptive statistics and generalized linear models based on analyses of covariance. All caregivers across various education levels reported noticing, reading, and understanding the poster (100%). Nearly all reported that the poster added new information to their knowledge (17 out of 21). In the additional analysis, the patients in the poster-exposed group showed greater improvements in the health outcomes, compared with the non-exposed group, although the effects were statistically non-significant within this study’s one-month period. A poster with pictorial information was effective in increasing awareness about dysphagia-specific information among caregivers of patients in inpatient care and can be used as an augmentative means of information, with potential benefits for patient safety and rehabilitation.

## 1. Introduction

Dysphagia is a swallowing disorder associated with difficulties in forming or safely moving food from the mouth into the stomach. It occurs following mechanical, neurological, or presbyphagia disorders and is particularly predominant among patients with neurological conditions, elderly people, and patients with head and/or neck diseases. Dysphagia can lead to clinically relevant complications such as malnutrition, dehydration, weight loss, pulmonary complications, pneumonia, and, in the most severe cases, death [1,2,3,4,5].

To avoid or reduce the occurrence of these complications and improve a patient’s health, careful eating procedures and food diet modifications are necessary as part of the treatment and rehabilitation of patients with dysphagia [4,6]. However, the inability to eat orally and without diet restrictions often interferes negatively with a patient’s emotional and psychosocial functioning, namely, through loss of pleasure associated with food and eating. The required dietary changes are sometimes perceived as abrupt, which compromises the transition to the desired food regimens and adherence, namely due to a faulty understanding of the reasons for the changes [6,7,8,9]. Implementing the dietary changes, including adapting the indicated food consistencies and observing contraindicated foods, can also represent a heavy daily burden for caregivers [5,6,7,10,11,12].

In inpatient care, these food procedures are carried out by trained health professionals [8,13,14]. Professionals trained in swallowing strategies have been associated with a much lower incidence of aspiration pneumonia among dysphagia patients, compared with family members who fed the patients without receiving this training [15]. While food-related procedures are crucial for a patient’s health and safety, caregivers (and patients) might find it difficult to observe them, especially if their understanding of the necessity for such procedures is limited [6,7,9].

Ideally, a specialized professional would explain the risks and consequences of dysphagia to caregivers, encourage their involvement in the intervention process, and use appropriate visual references to raise their awareness and help them understand important aspects associated with this swallowing difficulty, such as the texture of the food that is to be adapted [1,7,8,15]. However, research suggests that important communication failures exist between health professionals, the patients, and their relatives, regarding the necessity of these dysphagia-related feeding changes [7,8,13]. Studies indicate that a professional’s communication about the risks of dysphagia tends to be based on punitive and technical language that emphasizes the patient’s limitations, reduces the motivation to eat, and makes it difficult for the person to understand the whole process of diet change [6,7]. Both patients with dysphagia and their family members have reported the need for more personalized and practical information regarding dysphagia management. Some authors have claimed that education about dysphagia should be standardized and made widely available [1,5,7,8,12]. When oral communication is insufficient or difficult to understand, visual references can contribute to filling in the gaps or even be used in place of oral information [7,16,17,18,19].

Studies using visual elements for communicating risks and food procedures to caregivers of patients with dysphagia in inpatient care were not found in the literature. In a previous study on a caregiver’s knowledge about the recommendations taught by speech and language therapists regarding dysphagia management, visual aids, such as videofluoroscopy (i.e., a swallowing exam), were used to strengthen the impact of the training, namely, on the health risks and difficulties associated with dysphagia. However, the effect of this type of visual aid was not assessed [20]. In another study, a pictorial meal mat containing dysphagia descriptors was placed above the patient’s bed to improve the communication of new dysphagia recommendations among the staff. This visual way of conveying information was successful, namely, in drawing the staff’s attention to patients with dysphagia; the authors concluded that the meal mats ensured that dysphagia recommendations stood out. However, the target population in this study was the staff and not the caregivers of patients with dysphagia [21]. Likewise, in a different study about the barriers to compliance with dysphagia recommendations experienced by South African nurses, the authors suggested the development of supporting material, such as glossaries and visual aids, namely, to improve knowledge and communication among the staff regarding dysphagia [22].

Visual signals have also proven effective in other health areas, and warning symbols combined with education have been widely used in health care to promote safety-appropriate behaviors [16,17,18,19,23,24]. For example, in a study conducted in nursing homes, where patients often have dysphagia, caution symbols combined with education were effective in reducing the erroneous smashing of medication, which is a well-known and common problem in that context [16]. Standardized color-coded signs were used in another study aimed at increasing knowledge about infection prevention and compliance with isolation precautions among staff members in a hospital [18]. The signs increased team and patient safety, as well as knowledge about, and comfort with, infection-control practices and guidelines. The standardized color-coded isolation signs were visible and saved time by providing immediate information to professionals, who did not need to constantly refer to the actual infection of each patient [18]. In another study, the findings indicated that visual aids, more than verbal information, improved the capacity of individuals with dementia to comprehend medical information, employ supportive reasons, and relate this information to their own situation [19].

These studies show the belief that the use of visual aids can improve knowledge, communication, and compliance in situations of dysphagia. They have also shown the successful use of pictorial forms for conveying information among staff, both in the context of dysphagia and in other health contexts, and with patients in health areas other than dysphagia. They call for the need to develop visual aids that support care and rehabilitation in dysphagia. However, studies on these means of conveying information to caregivers of patients with dysphagia are lacking. It is possible that the use of visual signs communicating risks and food procedures to caregivers of patients with dysphagia also yields positive results. Studying such augmentative forms of conveying this information is necessary as a way to complement or support speech [19], especially in view of the gaps in patient-professional communication about dysphagia, the difficulties of caregivers in adhering to the changes and understanding their reasons, and a caregiver’s need for more knowledge on this swallowing disorder, as reported in research. If visual signs are effective then they can provide an important contribution to the safety and rehabilitation of patients with dysphagia.

The aim of this study was to assess the usefulness of a visual poster with communication symbols depicting dysphagia-related risks and food procedures from the point of view of caregivers of patients with dysphagia in inpatient care. The possible impact of the caregiver’s exposure to the poster on the patient’s dysphagia-related health was additionally explored. Even though dysphagia persists in many cases [25], improvements can occur, particularly among patients who are in the post-acute phase of the pathology [14,25]. If the poster is effective in informing caregivers about dysphagia-associated risks and food procedures, then some improvement might be observed in a patient’s dysphagia-related health as a result.

## 2. Materials and Methods

### 2.1. Sample

Participants were the caregivers of patients with dysphagia in medium-term and long-term inpatient health units in Porto, Portugal. They were invited to the study and received verbal and written information about its goals. Participation was voluntary, and a code was assigned to each case to ensure anonymity. Those who agreed to participate signed an informed consent form. This study received ethical approval (No. 082/2018) from the Ethics Committee of the Northern Health Regional Administration Area (ARSN), which is the government organization that oversees all healthcare units in the North of the country, and by the National Commission for Data Protection.

As the inclusion criteria, patients were eligible if the lesion associated with their dysphagia occurred at least six weeks prior (post-acute phase), if they remained in inpatient care for at least one month after the first moment of data collection, were 18 years or older, and received regular visits (e.g., weekly) from their family members or caregivers. Exclusion criteria were having caregivers who were foreigners or who had (uncorrected) vision problems that could interfere with poster visualization reported by the caregiver.

### 2.2. Instruments

Caregivers responded to a questionnaire about the poster communicating dysphagia-specific risks and food procedures one month after the poster was placed by the patient’s beds. The questionnaire asked whether the caregiver: (1) noticed the poster in the patient’s room; (2) read it; (3) found it useful, with the following response options: “I didn’t understand it”, “I understood it”, and “It was very important to me”; (4) considered that it added any new knowledge; and (4.1) which knowledge was added. Data on the caregiver’s relationship with the patient, visit frequency, age, gender, and education level were also obtained. Information on the patient’s age, gender, diagnosis, and time since admittance to inpatient care was additionally obtained from the patient’s clinical records.

The additional exploration of whether the poster affected the patient’s dysphagia-related health was based on the inspection of changes over time in the scores of several instruments that are used to assess dysphagia. Previous research indicates that the treatment of dysphagia should target three main risk factors associated with the occurrence of aspiration pneumonia, namely, reduced swallowing safety (causing the aspiration of pathogens into the patient’s airways), decreased nutritional status, and poor oral health and hygiene related with respiratory pathogens colonizing the mouth [2]. Accordingly, the Gugging Swallowing Screen (GUSS) assessing the risk of aspiration, the Functional Oral Intake Scale (FOIS), the Mini Nutritional Assessment (MNA), and the Oral Health Assessment Tool (OHAT) were used in this study.

Considered the most appropriate scale for assessing patients admitted to inpatient care, the GUSS assesses the risk of aspiration from dysphagia through separate evaluations for non-fluid and fluid nutrition [26]. It consists of two parts. Part 1 assesses the patient’s state of vigilance, presence of a cough, throat clearing, and capacity for swallowing saliva (divided into successful, drooling, and voice change). Each item receives a “Yes” or “No” score. Part 2 evaluates the patient’s swallowing capacity directly for liquids, semisolid food, and solid food, respectively, as “not possible” (corresponding to 0), “delayed” (corresponding to 1), and “successful” (corresponding to 2). The swallowing of liquids, semisolid food, and solid food, each, is evaluated for the presence of a cough, drooling, and voice change (with “Yes” or “No” scores given in each case). The scale’s total is the sum of the points obtained in Part 1 with the points obtained in Part 2. The maximum score is 20 points, indicating an adequate swallowing capacity without a risk of aspiration. The minimum score is 0 points, indicating severe dysphagia with a high risk of aspiration [26]. Good psychometric properties have been reported for this scale [26].

The FOIS presents seven classification levels that reflect the patient’s capacity for oral intake, thus providing an effective indication of the degree of severity of the dysphagia. The scale’s values range from 1 (indicating no oral intake) to 7 (indicating total oral intake without restrictions), with various degrees of tube-feeding dependency and diet modifications between these values. A high inter-rater reliability level has been reported for this scale, with perfect agreement on 85% of the ratings. Kappa statistics ranging from 0.86 to 0.91, and a high consensual validity (of 0.90) have also been reported [27].

The MNA is one of the most frequently used and recommended scales for nutritional assessment in all health areas, namely due to its ability to identify the risk of malnutrition. The Mini Nutritional Assessment Short-Form (MNA-SF), used in this study, comprises the following six items: diminished food intake (due to loss of appetite, digestive problems, and chewing or swallowing difficulties), weight loss, body mass index, mobility, experience of stress or acute disease in the last three months, and neuropsychological problems [28]. Scoring varies for each item (e.g., from 0 to 3 or from 0 to 2) and the final score is the sum of all the aspects [28]. The MNA-SF’s maximum score is 14, corresponding to an adequate nutritional state. Its minimum score is 0, corresponding to the presence of malnutrition. The MNA-SF has shown sensitivity and predictive values of 75.4% and 79.9%, respectively [28].

The OHAT is a simple screening tool that assesses the health of the oral cavity in eight categories (lips, tongue, gums and tissues, saliva, natural teeth, dentures, oral cleanliness, and dental pain). Each aspect is measured on a 0- to 2-point Likert scale. The scores obtained for each of the eight categories are added to a total score. The minimum possible score is 0, corresponding to a healthy state. The maximum possible score is 16, corresponding to an unhealthy state of the oral cavity. The content validity of the OHAT has been established in various studies. For inter-rater reliability, percentage agreements have ranged from 72.6% for oral cleanliness to 92.6% for dental pain. The Kappa statistic has shown moderate values (ranging from 0.48 to 0.60) for lips, tongue, gums, saliva, and oral cleanliness. For all other categories, the Kappa statistic has ranged from 0.61 to 0.80, indicating substantial agreement [29].

### 2.3. Procedures

The poster was created for the purposes of this study following the principles of design. Visual posters communicating information through signs can be particularly effective if they abide by the principles of design. These include the use of simple elements that contain little detail, are often familiar, and maintain semantic relationships among them (i.e., as if “telling a story”) [23,24]. The combination of colors, words, and symbols provides variations in the levels of the reported danger (e.g., red, orange, and yellow generally indicate decreasing levels of danger) [23]. A triangle containing an exclamation point can further reinforce the level of risk conveyed, indicating the risk of personal injury [18,23]. An art designer was enlisted in this study to project and create the poster in collaboration with speech and language therapists.

The poster stood 29.7 cm by 21.0 cm in size and contained a title (“Guidelines for the family member/caregiver”), a subtitle (“Patient with food difficulties”), a warning sign (a triangle with an exclamation point), and 12 images, including human figures, depicting the dangers and the procedures associated with feeding the patient with dysphagia (Figure 1). The images were organized in three rows referring, respectively, to risks (first row identified in red, with four images), adequate food procedures (second row identified in yellow, with five images), and necessary food adaptations (third row identified in green, with one and/or two images). The third row referred to different diets and liquid consistencies appearing in self-adhesive papers that health professionals posted on site, according to each patient’s needs. A brief description of each image also appeared below the respective image.

Prior to its placement by the patients’ beds, the poster was validated by a panel of nine people that included both experts in dysphagia (e.g., speech and language therapists) and lay people who were ignorant of dysphagia. Their comments were collected via an online questionnaire containing several closed- and open-ended questions that asked for their opinion, namely, considering the target population of the patient’s caregivers or family members (e.g., “In your opinion, only the images (without the words) in the poster are easy to understand?”, “Do you consider the words together with the images easy to understand?”, “What are the difficulties (if answering “no” to any previous questions)?”, “What improvements do you suggest?”). A final open-ended question asked, “What other changes would you make to the poster in terms of Colors, Size/Font, Organization, Structure?”. All their suggestions were compiled, discussed among the researchers and the designer, and integrated into the final version of the poster.

This final version of the poster was placed by the beds of the eligible patients for one month. In the end, the caregivers of these patients responded to the individual questionnaire about the poster in a separate room with adequate conditions of privacy.

For the additional goal of exploring whether the poster might affect a patient’s dysphagia-related health, the two-bed bedrooms of the 42 eligible patients who were receiving treatment in inpatient care were randomly assigned to control and experimental groups before the beginning of the study. For the randomization process, each room was assigned a number and allocated to the respective group through the use of a random number generator. Then, the caregivers of the patients in the intervention group were exposed to the poster that was placed by the beds for one month. In the comparison group, the caregivers were not exposed to the poster. Otherwise, nurses and the health team were the same for all the patients, including in the control and experimental groups, and the inpatient health units maintained their regular functioning, as before. Figure 2 depicts this study’s enrollment flow chart. A patient’s dysphagia-related health status in both the intervention and comparison groups was assessed before the placement of the poster and again one month later. The assessment was conducted by the same speech and language therapist, who ensured the consistency of procedures, evaluation parameters, and criteria across all patients.

### 2.4. Analyses

The descriptive statistics consisted of absolute and relative frequencies for qualitative variables, means and standard deviations for quantitative variables, or medians and interquartile ranges for quantitative variables that failed to meet normality assumptions in the preliminary analyses with the Shapiro–Wilk’s test. To inspect any differences between control and experimental groups regarding socio-demographic characteristics, *χ*^2^ tests for qualitative variables, *t*-tests for parametric quantitative variables, and Mann–Whitney *U* tests for non-parametric quantitative variables were performed with the level of statistical significance set for less than 0.05. For the open-ended question that was asked at the end of the questionnaire, i.e., (4.1) which knowledge the poster added, the themes that caregivers brought up were identified, grouped by similarity, and counted, following the procedures of content analysis [30]. This coding and categorization process was fairly straightforward and yielded no points of disagreement between the two independent coders.

To assess the eventual effects that the poster might have on changes in a patient’s dysphagia-related health from Time 1 (T1) to Time 2 (T2), the difference between the scores at T2 and at T1 was calculated for each outcome (T2-T1). For the OHAT, T1 minus T2 was used so that its results could be read in the same direction as those of the other three outcomes. To control for baseline differences that were detected in preliminary analyses with *t*-tests or Mann–Whitney tests, as appropriate, between experimental and comparison groups on the outcome variables, the patient’s scores on the GUSS, FOIS, and OHAT were examined with the analysis of covariance (ANCOVA) procedures. The ANCOVA analyses were conducted through Generalized Linear Models, with Identity as the link function, given the observed non-normality of the T2-T1 differences (Shapiro–Wilk’s tests showing *p*-values less than 0.001 for the T2-T1 differences on all four outcome measures). In this study, the number of covariates was approximately within the recommended ratio (<0.10) of the total data for an analysis of covariance [31]. In each model, the difference in the patient’s scores at T1 and T2 was the response variable, poster exposure (group) was entered as the factor (categorized as comparison group = 0 and poster group = 1), and the baseline score at T1 (on the GUSS, FOIS, and OHAT) was entered as the covariate in the respective model. Because no baseline differences existed on the MNA, its score at T1 was not entered as a covariate in its respective model. The correlations between the baseline scores and the respective response variable were otherwise statistically non-significant (except for the OHAT, and the patient’s baseline scores at T1 were entered as a covariate, as mentioned above) [32].

Data were analyzed in IBM SPSS Statistics 28. The level of statistical significance was set at less than 0.05.

## 3. Results

Table 1 shows the characteristics of the participants. In regard to the various socio-demographic aspects studied, those who were exposed to the poster (*n* = 21) did not differ significantly from the remaining eligible participants in inpatient care who were not exposed (*n* = 21). Considering the entire sample as a whole, all caregivers were the patient’s family members and primarily were the patient’s descendants (*n* = 32, or 76.2%), especially their children (*n* = 25, or 59.5%). About 45% visited the patients on a daily basis, and just over half (54.8%) visited the patient a few times per week. The caregivers’ mean age was about 55 years old (*SD* = 13.24) and the proportions of men (47.6%) and women (52.4%) were similar. They had varied levels of education, ranging from less than four years of school to college degrees, with a greater concentration on six, nine, and 12 years of school (21.4%, 21.4%, and 26.2%, respectively).

Of the 42 patients who were eligible to participate, most were women (61.9%) and their ages ranged between 39 and 97 years old (mean = 77.26). A stroke was the most frequent diagnosis (59.5%), followed by several other diagnoses, such as traumatic brain injury (*n* = 3), femoral neck fracture (*n* = 2), hetero-aggression (*n* = 1), thalamic hemorrhage (*n* = 1), meningoencephalitis (*n* = 1), Parkinson’s disease (*n* = 1), Alzheimer’s disease (*n* = 1), dementia (*n* = 1), multiple sclerosis (*n* = 1), Whipple’s disease (*n* = 1), peripheral vascular disease (*n* = 1), disuse myopathy (*n* = 1), aspiration pneumonia (*n* = 1), and type-2 diabetes (*n* = 1). This diagnosis variable was dichotomized (into stroke vs. others) reflecting the concentration of the patients in the “stroke” category and the remaining patients’ dispersion across a variety of different diagnoses. A distinction between patients admitted to medium- and long-term inpatient care (i.e., institutional regime) was also used to reflect the improvement potential (assumed to be greater in medium-term inpatient care) regardless of the patient’s age or associated disease. Most patients were admitted to long-term care (76.2%), whereas 23.8% were admitted to medium-term care. The mean time in inpatient care was 1.48 years (*SD* = 1.23). No statistically significant differences were found between the poster and the comparison groups regarding any of these characteristics (i.e., age, gender, type of diagnosis, institutional regime, or years in inpatient care).

### 3.1. Caregivers’ Views of the Poster

All caregivers who were exposed to the poster responded to the questionnaire (*n* = 21). All reported that they (1) noticed the poster in the room, (2) read it, and (3) understood it, with three caregivers further indicating that they found it very useful. The vast majority (*n* = 17) considered that the poster (4) added knowledge that they had not received before. The new information (4.1) referred especially to risks (the first row in the poster) and adequate food procedures (the second row in the poster). In regard to the risks, caregivers indicated that the new information received from the poster included the possibility of food going into the patient’s lungs (for 5 out of the 17), the possibility of leading to pneumonia (for another 2), and the possibility of death (for 8 out of the 17), with others referring the risk of extreme thinness (another 2). In regard to adequate food procedures, caregivers referred to the image showing the healthcare professional correctly feeding the patient and mentioned the posture while eating as new knowledge received from the poster (7 out of the 17). A few (2 out of the 17) mentioned the adaptation of food preparation (the third row in the poster) and pointed specifically to liquid consistency as new information received from the poster. Several caregivers reported more than one category of new information received from the poster (*n* = 7), namely, “pneumonia and death”, “fluid consistency and death”, or “extreme thinness, feeding posture, and 90° bed position”.

The four caregivers who reported that the poster added no new information had varied family relations with the patients, and varied ages, genders, and education levels. Two of them were the patient’s sons, being 65 and 31 years of age, and with 4 and 12 years of school, respectively. The other two were the patient’s sister and granddaughter, who were 43 and 33 years old, and with 12 years of school and a college degree, respectively. Three had their patients in long-term care units and one had their patient in a medium-term care unit.

### 3.2. Impact of the Poster on Patients’ Dysphagia-Related Health Outcomes

If the poster effectively changed the caregivers’ awareness of the risks involved in dysphagia and their feeding behaviors, then, as a result, greater improvements in the patients’ dysphagia-related health might be observed in the group that was exposed to the poster when compared with the other (non-exposed) group. If the latter (non-exposed) group, which was equivalent to the exposed group in all aspects (through the randomization procedure) except for exposure to the poster, would register comparatively fewer positive changes in the patients’ dysphagia-related health, then this provided evidence of the poster’s effectiveness.

Figure 3 depicts the differences in scores after a one-month period (between Time 1 and Time 2) for the four dysphagia-related health outcomes by group. Greater changes correspond to greater health improvements.

The intervention group registered greater improvements than the comparison group in regard to all four dysphagia-related health outcomes. Table 2 shows the mean improvements by group and the results of the regression models. The effect of the poster did not reach statistical significance across all four dysphagia-related health outcomes, though, assuming increasingly greater *p*-values, respectively, for the MNA, *β* = 0.476, *p* = 0.064; GUSS, *β* = 0.508, *p* = 0.089; FOIS, *β* = 0.285, *p* = 0.297; and OHAT, *β* = 0.156, *p* = 0.635 (Table 2).

## 4. Discussion

The results of this study indicate that the poster had a positive influence overall. Designed as an augmentative means of information for caregivers of patients with dysphagia, the poster communicated dysphagia-specific risks and food procedures through visual signs and images accompanied by a few written words. It followed the recommendations for the construction of this type of figurative content and showed effectiveness, including in drawing a caregiver’s attention to it, namely, through the use of a warning sign placed at the beginning, which is a well-known sign of danger [18,23,24]. Across a varied range of education levels (from four years of school or less to college degrees), all caregivers who were exposed to the poster (100%) reported that not only did they notice the poster, but they also “read” and understood it. Thus, these findings indicate that this type of visual aid had similarly positive results with caregivers of patients with dysphagia as it had with ward staff in a previous study that focused on the communication of new dysphagia recommendations through a pictorial meal mat placed above the patients’ beds in the hospital [21].

The fact that nearly all the caregivers (i.e., 17 out of 21) reported that the poster added new information to their knowledge about dysphagia reinforces its usefulness. No pattern was discernible among the remaining four caregivers, who reported that the poster added no new information, namely, regarding gender, age, education level, or family relation with the patient.

Among the group of caregivers reporting that the poster added new information, the new information that was learned pertained more to risks and feeding postures than to the type of diet and consistencies of solid food and liquids. A previous study indicated that caregivers remember especially tangible and routinely used strategies of managing dysphagia learned from speech and language therapists, such as food and drink consistency [20], which could explain these results, although, feeding postures could arguably be as tangible and as routinely used. It may be that, in this sample, any previous knowledge received about dysphagia focused more on the diet and on food and drink consistency, and less on risks and adequate feeding positions. This is consistent with previous research on the provision of information in hospital discharge communications, reporting nearly double omissions for postural recommendations, compared with diet recommendations [8]. It is also possible that the choice of colors (risks identified in red, adequate procedures in yellow, and type of diet and food consistency in green), and the relative position of each type of information in the poster (vertically from top down) might have drawn the caregivers’ special attention to the red and yellow, which are associated with (decreasing) levels of danger, and less to the green row. The caregivers’ attention could also focus more on the upper (initial) rows than on the row positioned at the bottom (the last row) [23]. Other studies can be useful to illuminate the influence that colors and the relative position of the information might have on the caregivers’ perceptions. Further, more studies are necessary to explore the extent of a caregiver’s actual knowledge of dysphagia, as well as its sources.

Overall, the poster, which was previously validated by a panel of experts and laypeople, had positive results in terms of readability (including understandability) and usefulness among its target population, i.e., caregivers of patients with dysphagia across a range of education levels. Thus, this study adds caregivers of patients with dysphagia to previous research also reporting positive uses of visual signs among staff, including in the context of dysphagia, and among patients in other health areas [16,17,18,19,21,23,24]. Caregivers express the desire for more knowledge about dysphagia procedures and report that communication of this information is often limited or delivered with technical language [7,8,13]. The poster can play an important role in bridging these communication gaps. Furthermore, its fixed posting by the patients’ beds may serve the double purpose of providing important information initially and working as a continuous reminder of that information afterward, for incoming visitors.

In regard to the effect of the poster on a patient’s dysphagia-related health outcomes, the fact that greater improvements occurred in the poster-exposed group when compared with the non-exposed group was promising, especially within the short period of time considered in this study (one month). The causal role of the poster on these improvements is based on the randomized controlled design coupled with the adjustment for baseline differences used in this study. With these procedures, the improvements in the risk of aspiration associated with non-fluid and fluid nutrition accompanied improvements in the degree of dysphagia, the health of the oral cavity, and the patient’s nutritional status in the poster group. Whereas, in the comparison group, improvements in these outcomes were comparatively smaller (except for the health of the oral cavity) and were null for the patient’s nutritional status.

The fact that, among all four outcomes, the health of the oral cavity (assessed with the OHAT) was the one outcome for which the two study groups differed the least from each other is possibly related to the treatment of this aspect in inpatient care, which falls greatly under the healthcare unit’s responsibility, instead of the caregiver’s intervention. The poster also contained no information specifically directed at this aspect. At the other end of the spectrum, the greatest difference observed between the two study groups occurred as regards nutritional status (assessed with the MNA), possibly as a result of the improvements in a patient’s dysphagia condition, namely, due to adherence to the indicated food procedures, including feeding postures, types of diets, and food and drink consistencies, all of which were depicted in the poster. The fact that the overall pattern of improvements on the GUSS and on the FOIS was similar was an unsurprising result because these two scales assess closely related aspects, namely, the risk of aspiration associated with a certain degree of dysphagia, and food swallowing capacity, respectively. One result actually validates the other.

Overall, the results show a coherent pattern of greater dysphagia-related health improvements for the poster-exposed group, compared with the non-exposed group. Nevertheless, the effects of the poster were statistically non-significant across all four outcomes. This might have to do with the limitations associated with the length of time considered and with the sample size. More than one month may be necessary for considerable improvements to occur in a patient’s aspiration risk associated with fluid and non-fluid nutrition, levels of dysphagia, nutritional status, and oral hygiene. One month was used in this study because many patients leave inpatient care after this time, which would limit the sample size and composition. The strict inclusion criterion of patients receiving regular visits from their caregivers also limited the number of participants. Despite the inclusion of caregivers of patients receiving treatment in both medium- and long-term health units, greater changes are expected to occur in the medium-term than in long-term care. The small proportion of patients in medium-term health units probably contributed to the observed non-significant differences (an additional graphical inspection revealed that dysphagia-health improvements did occur among the patients in medium-term health units in all four outcomes, not among the patients in long-term care). Finally, although the assessment at T1 occurred before the sample’s randomization, the poster could not be made blind at T2, which could result in an expectation bias and, due to the small number of participants, sampling bias also cannot be completely ruled out. Still, the effects of the poster might be statistically significant over more prolonged periods of time and with larger samples, which can be examined in future studies with randomized controlled designs and fully blinded procedures.

The poster’s usefulness and impact can now be inspected among other samples in future studies, namely, from inpatient care units across different geographic regions, or from outpatient care, by providing caregivers with the poster to help them deliver care at home. Additionally, to help inter-professional communication, posters conveying information about dysphagia through signs and images can also be designed for multidisciplinary teams, potentially including more detailed information (e.g., maneuvers facilitating ingestion) than they usually contain when they are designed for patients or their caregivers.

## 5. Conclusions

A poster communicating information through signs and images was effective in increasing awareness about dysphagia-specific information among caregivers of patients followed in inpatient care, with potential benefits for the patient’s health. Based on these results, the use of the poster is recommended as a visual means of conveying information to caregivers of patients with dysphagia, helping to bridge possible communication gaps between the multidisciplinary team and the caregivers. It is an economic means of communication with potential benefits for family well-being and for patient safety, rehabilitation, and health-related expenses that could be widely implemented in inpatient care settings in the future.

## Figures and Tables

**Figure 1 healthcare-12-00148-f001:**
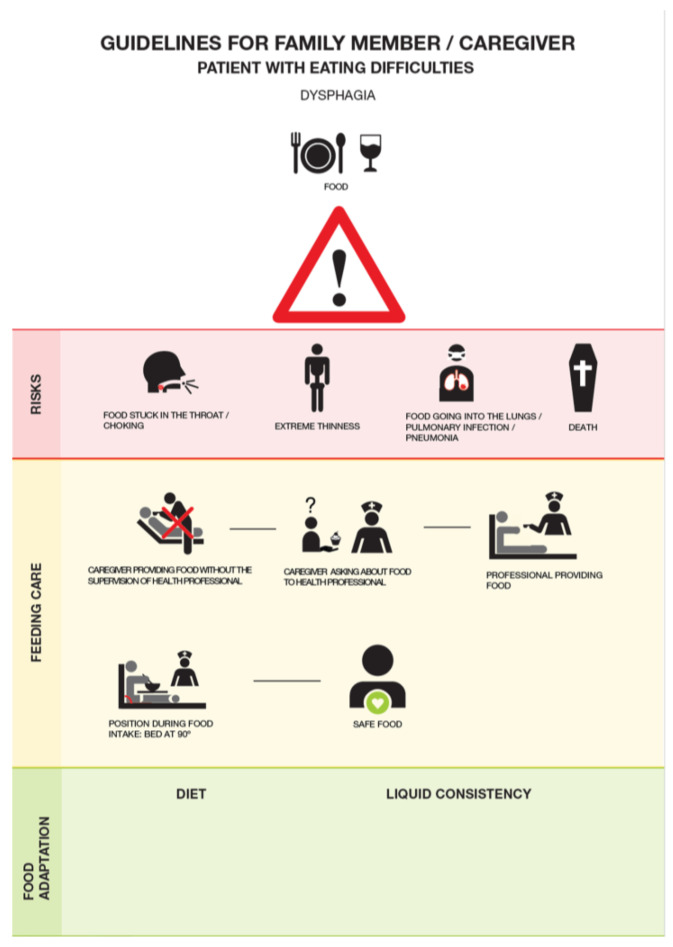
Poster communicating information on dysphagia-associated risks and food procedures.

**Figure 2 healthcare-12-00148-f002:**
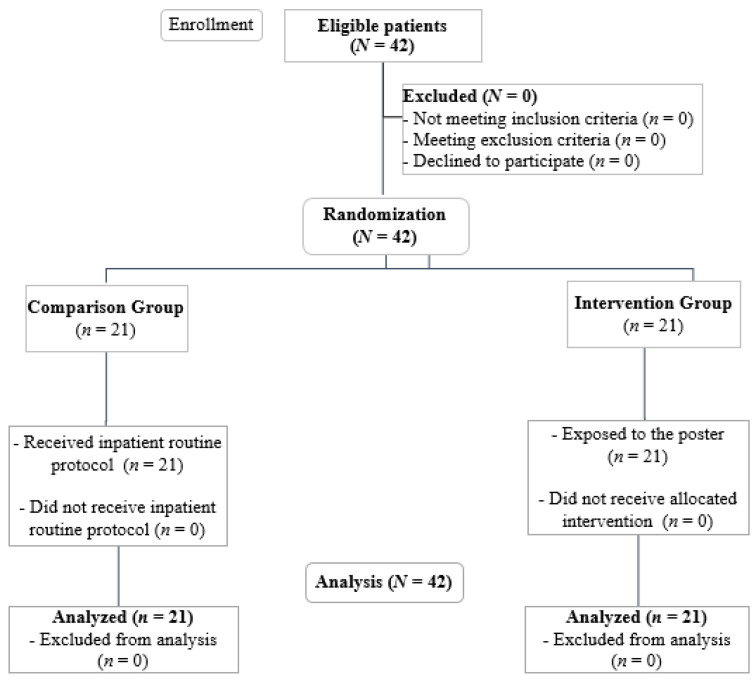
Flow diagram of enrollment, eligibility screening, and analysis.

**Figure 3 healthcare-12-00148-f003:**
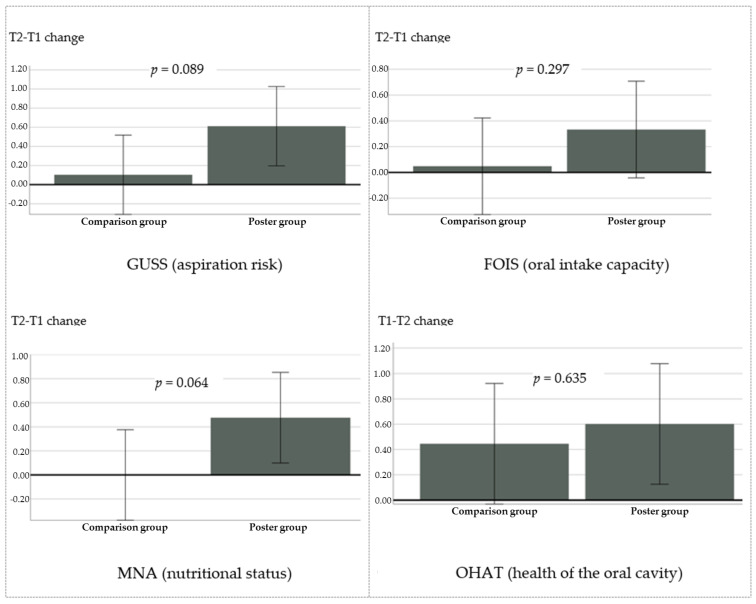
Estimated marginal means and 95% confidence intervals of the differences between Time 1 (T1) and Time 2 (T2) in dysphagia-related health outcomes over a one-month period by group, adjusting for baseline differences.

**Table 1 healthcare-12-00148-t001:** Sample characteristics (*N* = 42) and differences between poster and comparison groups.

		Poster Group		Comparison Group		Total Sample		Difference *	
		*n* = 21		*n* = 21		*N* = 42		Test	*p*
Caregivers									
Relationship with the patient—*n* (*%*)								-	-
	Children	15	(71.4)	10	(47.6)	25	(59.5)		
	Grandchildren	1	(4.8)	1	(4.8)	2	(4.8)		
	Nieces/Nephews	2	(9.5)	3	(14.3)	5	(11.9)		
	Spouses	1	(4.8)	5	(23.8)	6	(14.3)		
	Siblings	1	(4.8)	1	(4.8)	2	(4.8)		
	Parents	1	(4.8)	1	(4.8)	2	(4.8)		
Visit frequency—*n* (*%*)								2.403 ^a^	0.121
	Daily	12	(57.1)	7	(33.3)	19	(45.2)		
	A few times per week	9	(42.9)	14	(66.7)	23	(54.8)		
Age—mean (*SD*)		53.38	(12.59)	57.29	(13.89)	55.33	(13.24)	0.955 ^b^	0.346
Gender—*n* (*%*)								0.382 ^a^	0.537
	Female	12	(57.1)	10	(47.6)	22	(52.4)		
	Male	9	(42.9)	11	(52.4)	20	(47.6)		
Education level—*n* (*%*)								-	-
	Illiterate	0	0	1	4.8	1	2.4		
	1st year	1	4.8	2	9.5	3	7.1		
	4th year	2	9.5	4	19	6	14.3		
	6th year	4	19	5	23.8	9	21.4		
	9th year	5	23.8	4	19	9	21.4		
	12th year	6	28.6	5	23.8	11	26.2		
	College degree	3	14.3	0	0	3	7.1		
Patients									
Age—median (*IQR*)		84.00	(66.00–86.50)	78.00	(69.00–86.00)	80.50	(68.00–86.25)	−0.012 ^c^	0.991
Gender—*n* (*%*)								0.000 ^a^	1.000
	Female	13	(61.9)	13	(61.9)	26	(61.9)		
	Male	8	(38.1)	8	(38.1)	16	(38.1)		
Diagnosis—*n* (*%*)								0.099 ^a^	0.753
	Stroke	12	(57.1)	13	(61.9)	25	(59.5)		
	Other	9	(42.9)	8	(38.1)	17	(40.5)		
Institutional regime—*n* (*%*)								0.525 ^a^	0.469
	Medium-term care	6	(28.6)	4	(19.0)	10	(23.8)		
	Long-term care	15	(71.4)	17	(81.0)	32	(76.2)		
Years in inpatient care—median (*IQR*)		0.88	(0.36–2.07)	0.98	(0.32–2.98)	0.96	(0.21–2.28)	215.500 ^c^	0.900

Note. *IQR*—interquartile range. *SD*—Standard deviation. * Difference between intervention and comparison groups: Test— ^a^ *χ*^2^ test for dichotomous variables, ^b^ *t*-test or ^c^ Mann–Whitney *U* test for continuous variables. Dashes mean that a test was not computed when its assumptions were not met.

**Table 2 healthcare-12-00148-t002:** Estimated marginal means (and standard errors) of improvements in the patients’ dysphagia-related health scores after a one-month period for the poster group and for the comparison group, adjusting for baseline differences.

	Estimated Marginal Means (Standard Errors)		Generalized Linear Models ^a^		
	Comparison Group (*n* = 21)	Poster Group (*n* = 21)	*β*	Wald *χ*^2^	*p*
GUSS ^1^ change	0.10	0.61	0.508	2.890	0.089
	(0.20)	(0.20)			
FOIS ^2^ change	0.05	0.33	0.285	1.090	0.297
	(0.18)	(0.18)			
MNA ^3^ change	0.00	0.48	0.476	3.420	0.064
	(0.18)	(0.18)			
OHAT ^4^ change	0.45	0.60	0.156	0.225	0.635
	(0.23)	(0.23)			

^a^ Test (for 1 degree of freedom) with 1—poster group (vs. 0—comparison group). ^1^ GUSS = risk of aspiration associated with fluid and non-fluid nutrition, ^2^ FOIS = oral intake capacity/dysphagia severity, ^3^ MNA = nutritional status, ^4^ OHAT = health of the oral cavity. Greater changes correspond to greater health improvements.

## Data Availability

Data described in the manuscript will be made available upon request pending application and approval.

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
