# Peer review of "Communicating Risks and Food Procedures through a Visual Poster for Caregivers of Patients with Dysphagia in Inpatient Care: Usability and Impact"

_healthcare, 2024, doi:10.3390/healthcare12020148_

Round 1

Reviewer 1 Report

Comments and Suggestions for Authors

The sample size of this study was relatively small, with only 42 officers participating in the study. This is a significant limitation to the findings of the study. A larger sample size would have provided stronger evidence of the effectiveness of the posters. Secondly, the study only evaluated the impact of the posters within one month. A more comprehensive understanding of the long-term effects of posters on dysphagia-related health outcomes would be possible with a longer follow-up period.

Some other suggestions include

 1. How does this study control for nurse interference? As nurse competence may have an impact on the results of the experiment?

 2. It is recommended that the literature review be extended to include a more comprehensive analysis of existing research on dysphagia and the use of visual aids in healthcare settings. This would enhance the relevance and significance of the findings by providing a broader context for the study.

 3. Authors are encouraged to critically evaluate both the strengths and limitations of previous studies on dysphagia and the use of visual aids. This should include identification of gaps in the existing literature and how the current study can potentially contribute to filling these gaps.

 4. Authors must provide a comparative analysis between their findings and those of previous studies.

 5. Based on the results of the study, the authors are encouraged to provide practical recommendations tailored to health professionals and policy makers.

Author Response

Reviewer 1

The sample size of this study was relatively small, with only 42 officers participating in the study. This is a significant limitation to the findings of the study. A larger sample size would have provided stronger evidence of the effectiveness of the posters. Secondly, the study only evaluated the impact of the posters within one month. A more comprehensive understanding of the long-term effects of posters on dysphagia-related health outcomes would be possible with a longer follow-up period.

R.: Thank you for your revision and for helping us improve the paper. Below are the point-by-point responses to the comments. The corresponding changes in the manuscript are marked with “track changes” (and yellow color).

Indeed the sample size and time period were presented in the manuscript as potential limitations dictated by the need to exclude many patients who did not have caregivers’ visits (given the paper’s focus on the caregivers) and also to minimize the loss to follow-up after a one-month period, as described in the manuscript. We have included the word “limitations” in that part of the text to make this point clearer.

We agree in that a larger sample and a longer follow-up period could have provided stronger evidence of the poster’s effectiveness on the patients’ dysphagia-related health, as stated in the Discussion (toward the end of the manuscript). Nevertheless, the poster proved to be useful from the caregivers’ standpoint (which was the paper’s main focus), and the additional analysis that was conducted on its possible effects on patients’ dysphagia-related health showed promising results.

Some other suggestions include

  1. How does this study control for nurse interference? As nurse competence may have an impact on the results of the experiment?

R.: Nurses (and the health team) were the same for all the patients. They continued to provide regular care to all the patients and their caregivers, as usual, throughout the study’s period. This information is now made clear toward the end of the Procedure, thank you for the observation.

  1. It is recommended that the literature review be extended to include a more comprehensive analysis of existing research on dysphagia and the use of visual aids in healthcare settings. This would enhance the relevance and significance of the findings by providing a broader context for the study.

R.: We have now included more articles on dysphagia and the use of visual aids in healthcare settings to broaden the context for the study, as suggested (Introduction, 5th paragraph).

  1. Authors are encouraged to critically evaluate both the strengths and limitations of previous studies on dysphagia and the use of visual aids. This should include identification of gaps in the existing literature and how the current study can potentially contribute to filling these gaps.

R.: Following this suggestion, we have critically evaluated previous research on dysphagia and the use of visual aids, identifying gaps and explicitly stating the potential contribution of this study in view of these gaps (Introduction, 7th paragraph).

  1. Authors must provide a comparative analysis between their findings and those of previous studies.

R.: Comparative analyses between the findings in this and in previous studies were included in the Discussion (1st paragraph, 3rd and 4th paragraphs).

  1. Based on the results of the study, the authors are encouraged to provide practical recommendations tailored to health professionals and policy makers.

R.: The recommendation on the importance of using the poster for caregivers of patients with dysphagia, based on the results of this study, was made explicit in the Conclusion, thank you for this suggestion.

Reviewer 2 Report

Comments and Suggestions for Authors

Dear authors

Thank you for submitting your draft titled "Communicating risks and food procedures through a visual poster for caregivers of patients with dysphagia in inpatient care: Usability and impact." 

My main concern are releted with the  definition of inclusion/exclusion criteria. It is suggested strengthen  your statistical strategy  according to the suggested recommendations to reduce the probability of bias and  type I and type II errors.

Annexed in the .pdf document my specific comments and recommendations.

We invite you to take into consideration our comments and suggestions in the attached  .pdf to materialize  substantial improvements and resubmit them for consideration for publication.

Author Response

Reviewer 2

Dear authors

Thank you for submitting your draft titled "Communicating risks and food procedures through a visual poster for caregivers of patients with dysphagia in inpatient care: Usability and impact." 

My main concern are releted with the definition of inclusion/exclusion criteria. It is suggested strengthen your statistical strategy according to the suggested recommendations to reduce the probability of bias and type I and type II errors.

Annexed in the .pdf document my specific comments and recommendations.

We invite you to take into consideration our comments and suggestions in the attached .pdf to materialize  substantial improvements and resubmit them for consideration for publication.

R.: Thank you for your helpful comments and suggestions. We have addressed them all and have marked the corresponding changes in the manuscript with “track changes” (and yellow color). Below are the point-by-point responses to the comments.

Please state in the manuscript that these are the inclusion criteria.

Patients were eligible if the lesion associated with their dysphagia occurred at least six weeks prior (post-acute phase), if they remained in inpatient care for at least one month after the first moment of data collection, were 18 years or older, and received regular visits (e.g., weekly) from their family members or caregivers.

R.: The specification about these being the inclusion criteria was added in the manuscript, as requested.

I consider that it is important to define specific exclusion criteria beyond not meeting the inclusion criteria, since they take for granted many characteristics of their very defined population, which are not evident to readers. For example, the patient must have a minimum level of education that allows him to read and write, that he does not have visual disability due to any chronic disease such as diabetes and hypertension, that he does not have myopia, and if so, that he has glasses for correction and visual acuity 202/20, who speaks the native Portuguese language, if he is an immigrant or foreigner, who has a minimum level of Portuguese (B1?) etc...

There may be a selection bias if the inclusion and exclusion criteria of the study individuals are defined in detail...

R.: Because the poster was for the caregivers, not for the patients, no other exclusion criteria were applied to the patients. However, we now have specified that exclusion criteria consisted of having caregivers who were foreigners or who had (uncorrected) vision problems that could interfere with poster visualization, reported by the caregiver. The fact that no participant was excluded based on the exclusion criteria is now also presented in the enrollment flow diagram (Figure 2). These changes were made in the post-revised version of the manuscript.

The issue of education/literacy was very important to us because many caregivers in Portugal have low literacy/education levels. The poster was created to be accessible also to people with low literacy levels and was pre-tested with illiterate people among the panel of experts and lay people. Thus, the presence of various education levels in this sample was an asset (not an exclusion criterion) - as described in the Discussion (4th paragraph).

Analyses. Use Pearson chi square to estimate differences between qualitative variables? P-value less 0.05? Please define…

R.: These procedures for the descriptive analyses were now defined in the Analyses section (1st paragraph), as suggested.

Also, the word “Quantitative” was removed from the last paragraph of the Analyses section, following the indicated suggestion.

Randomization. Please detail in the procedures section the randomization process of the individuals and what procedure was used.

R.: More information on the randomization process was included in the Procedures (last paragraph), following this suggestion.

Note to Table 1. Please add this information with more detail in analysis section.

R.: The information that is in the note to Table 1, on the statistical tests, was added with more detail in the Analysis section (1st paragraph), as suggested.

Discussion: Randomization. Please detail in the materials and methods section the randomization process of the individuals and what procedure was used.

R.: The randomization process was detailed in the materials and methods section, following this suggestion (in the Procedures, 5th paragraph).

Reviewer 3 Report

Comments and Suggestions for Authors

I believe it is worthwhile to communicate risks and eating procedures through visual posters for caregivers of patients with dysphagia in inpatient care. However, I am concerned about the following points, which led me to this view.

Number of subjects and statistical analysis

The number of subjects (42) is too small. Thus, sampling bias cannot be ruled out. Also, the fact that the age range of the subjects (39 and 97 years old) is lumped together is questionable. In addition, the following diseases were included in addition to stroke. traumatic brain injury (n = 3), femoral neck fracture (n = 2), heteroaggression (n = 1), thalamic hemorrhage (n = 1), meningoencephalitis (n = 1), Parkinson's disease (n = 1), and Parkinson's disease (n = 1), Alzheimer's disease (n = 1), de mentia (n = 1), multiple sclerosis (n = 1), Whipple's disease (n = 1), peripheral vascular disease (n = 1), disuse myopathy (n = 1), aspiration pneumonia (n = 1), and type-2 diabetes (n = 1). The pathogenesis of these diseases differs greatly. A detailed explanation is needed in this regard.

The authors have chosen Analysis of Covariance with Generalized Linear Models (ANCOVA) for the GUSS, FOIS, and OHAT patient scores to control for baseline differences detected in the preliminary analysis.

First, your explanation of the preliminary analysis is insufficient: in Table1. your mention t-test for continuous variables and χ2 test for dichotomous variables.

If so, I need to present the results of the normality test and add that the data are parametric. Note that in dichotomous variables, there are several cases where the expected value of a cell is less than 5. In this case, I believe that the Fisher's exact test should be selected instead of the X2 test.

Furthermore, it is said that the upper limit for the number of explanatory variables is about 1/15 of the total data for an analysis of covariance. This point also needs to be explained.

Without consideration of the above steps, it is difficult to determine the consistency of the authors' views.

That is all.

Please consider revising it.

Author Response

Reviewer 3

I believe it is worthwhile to communicate risks and eating procedures through visual posters for caregivers of patients with dysphagia in inpatient care. However, I am concerned about the following points, which led me to this view.

R.: Thank you for the useful comments and suggestions. We have addressed them marking the corresponding changes in the manuscript with “track changes” (and yellow color). Below are the point-by-point responses to the comments.

 Number of subjects and statistical analysis

 The number of subjects (42) is too small. Thus, sampling bias cannot be ruled out.

R.: Thank you for this observation. It has been included in the manuscript as a limitation, toward the end of the Discussion.

Also, the fact that the age range of the subjects (39 and 97 years old) is lumped together is questionable. In addition, the following diseases were included in addition to stroke. traumatic brain injury (n = 3), femoral neck fracture (n = 2), heteroaggression (n = 1), thalamic hemorrhage (n = 1), meningoencephalitis (n = 1), Parkinson's disease (n = 1), and Parkinson's disease (n = 1), Alzheimer's disease (n = 1), dementia (n = 1), multiple sclerosis (n = 1), Whipple's disease (n = 1), peripheral vascular disease (n = 1), disuse myopathy (n = 1), aspiration pneumonia (n = 1), and type-2 diabetes (n = 1). The pathogenesis of these diseases differs greatly. A detailed explanation is needed in this regard.

R.: Indeed, oropharyngeal dysphagia can occur at different ages and in a plethora of diseases. We distinguished between patients in medium- and in long-term care to reflect the potential improvement (assumed to be greater in medium-term care) regardless of age and of associated condition, and no significant differences were observed between the two groups under comparison (EG and CG) as regard either age or inpatient (medium- or long-term care) regime. We also used a dichotomized variable (stroke vs. others) reflecting the concentration of the patients in the stroke category with the remainder of the patients dispersed across a variety of other conditions. Using only patients with stroke would have limited the sample size greatly. Non-significant differences were found between the two (poster vs. comparison) groups as regard the type of disease (stroke vs. others). This information now appears under the Results (before Table 1), as well as the fact that the differences between EG and CG in all these and other socio-demographic variables in the study are statistically non-significant.

The authors have chosen Analysis of Covariance with Generalized Linear Models (ANCOVA) for the GUSS, FOIS, and OHAT patient scores to control for baseline differences detected in the preliminary analysis.

First, your explanation of the preliminary analysis is insufficient: in Table1. your mention t-test for continuous variables and χ2 test for dichotomous variables.

If so, I need to present the results of the normality test and add that the data are parametric. Note that in dichotomous variables, there are several cases where the expected value of a cell is less than 5. In this case, I believe that the Fisher's exact test should be selected instead of the X2 test.

R.: In response to this suggestion, the results in Table 1 were revised and now Table 1 includes small letters (a, b, c) in the “Test” column indicating the specific tests that were performed, for increased clarity. The note to Table 1 was also altered accordingly. In addition, more detailed information on these tests has been provided in the beginning of the Analyses section (1st paragraph), including the reference to the Shapiro Wilk’s test used in the preliminary analyses to assess normality. The Shapiro Wilk’s tests of normality showed p-values greater than 0.05 for caregivers’ age, both in the control (Shapiro Wilk statistic(21) = 0.960; p = 0.521) and experimental (Shapiro Wilk statistic(21) = 0.917; p = 0.076) groups, and patients’ age in the control group (Shapiro Wilk statistic(21) = 0.941; p = 0.230), with  p-values < 0.05 for patients’ age in the poster group (Shapiro Wilk statistic(21) = 0.892; p = 0.024) and years in inpatient care in both the poster group (Shapiro Wilk statistic(21) = 0.883; p = 0.017) and the comparison group (Shapiro Wilk statistic(21) = 0.859; p = 0.006). The fact that the differences between poster and comparison groups were all statistically non-significant regarding these and other socio-demographic characteristics was also added to the text in the Results section (end of 2nd paragraph).

A reference to these tests used in the preliminary analyses was also added to the beginning of the 2nd paragraph in the Analyses section (on the effects of the poster on changes in patients’ dysphagia-related status) from Time 1 to Time 2. The choice for Analysis of Covariance with Generalized Linear Models was made due to the ability to specify a non-normal distribution and non-constant variance as essential improvements of the generalized linear model over the general linear model, thus, the parametric and other assumptions (e.g., underlying linearity, or additive relationships) were not an issue in these analyses. This information was now explicitly stated in the Analyses section (middle of the second paragraph).

In the analyses with dichotomous variables, there was no case for which the expected value of a cell was less than 5, thus the Fisher’s exact test was not used or mentioned in the manuscript. The χ2 results were as follows:

Group * Visit_Freq Crosstabulation

Count 

Visit_Freq

Total

0

1

Group

0

7

14

21

1

12

9

21

Total

19

23

42

Chi-Square Tests

Value

df

Asymptotic Significance (2-sided)

Exact Sig. (2-sided)

Exact Sig. (1-sided)

Pearson Chi-Square

2,403a

1

,121

Continuity Correctionb

1,538

1

,215

Likelihood Ratio

2,427

1

,119

Fisher's Exact Test

,215

,107

Linear-by-Linear Association

2,346

1

,126

N of Valid Cases

42

a. 0 cells (,0%) have expected count less than 5. The minimum expected count is 9,50.

b. Computed only for a 2x2 table

Group * Caregiver_gender Crosstabulation

Count 

Caregiver_gender

Total

0

1

Group

0

10

11

21

1

12

9

21

Total

22

20

42

Chi-Square Tests

Value

df

Asymptotic Significance (2-sided)

Exact Sig. (2-sided)

Exact Sig. (1-sided)

Pearson Chi-Square

,382a

1

,537

Continuity Correctionb

,095

1

,757

Likelihood Ratio

,382

1

,536

Fisher's Exact Test

,758

,379

Linear-by-Linear Association

,373

1

,542

N of Valid Cases

42

a. 0 cells (,0%) have expected count less than 5. The minimum expected count is 10,00.

b. Computed only for a 2x2 table

Group * Patient_Gender Crosstabulation

Count 

Patient_Gender

Total

0

1

Group

0

13

8

21

1

13

8

21

Total

26

16

42

Chi-Square Tests

Value

df

Asymptotic Significance (2-sided)

Exact Sig. (2-sided)

Exact Sig. (1-sided)

Pearson Chi-Square

,000a

1

1,000

Continuity Correctionb

,000

1

1,000

Likelihood Ratio

,000

1

1,000

Fisher's Exact Test

1,000

,624

Linear-by-Linear Association

,000

1

1,000

N of Valid Cases

42

a. 0 cells (,0%) have expected count less than 5. The minimum expected count is 8,00.

b. Computed only for a 2x2 table

Group * Diagnosis_dic Crosstabulation

Count 

Diagnosis_dic

Total

0

1

Group

0

13

8

21

1

12

9

21

Total

25

17

42

Chi-Square Tests

Value

df

Asymptotic Significance (2-sided)

Exact Sig. (2-sided)

Exact Sig. (1-sided)

Pearson Chi-Square

,099a

1

,753

Continuity Correctionb

,000

1

1,000

Likelihood Ratio

,099

1

,753

Fisher's Exact Test

1,000

,500

Linear-by-Linear Association

,096

1

,756

N of Valid Cases

42

a. 0 cells (,0%) have expected count less than 5. The minimum expected count is 8,50.

b. Computed only for a 2x2 table

Group * Institut_regime Crosstabulation

Count 

Institut_regime

Total

0

1

Group

0

4

17

21

1

6

15

21

Total

10

32

42

Chi-Square Tests

Value

df

Asymptotic Significance (2-sided)

Exact Sig. (2-sided)

Exact Sig. (1-sided)

Pearson Chi-Square

,525a

1

,469

Continuity Correctionb

,131

1

,717

Likelihood Ratio

,528

1

,468

Fisher's Exact Test

,719

,359

Linear-by-Linear Association

,513

1

,474

N of Valid Cases

42

a. 0 cells (,0%) have expected count less than 5. The minimum expected count is 5,00.

b. Computed only for a 2x2 table

Furthermore, it is said that the upper limit for the number of explanatory variables is about 1/15 of the total data for an analysis of covariance. This point also needs to be explained.

R.: This point now appears in the Analyses section, around the middle of the second paragraph.

Without consideration of the above steps, it is difficult to determine the consistency of the authors' views.

That is all.

Please consider revising it.

R.: The paper was revised following the comments and suggestions above, thank you for your insights.

Round 2

Reviewer 1 Report

Comments and Suggestions for Authors

My gratitude goes to the author for the revisions done in this article. It was a pleasure to read this article. I can see that the author has worked hard to enhance this article and make it more valuable. This deserves acknowledgment.

Author Response

Thank you very much for the review and also for the nice words, we truly appreciated them.

Reviewer 3 Report

Comments and Suggestions for Authors

Thank you for your prompt and courteous corrections. 

I have read the revised draft.

I would like you to reconsider the authors' discussion following the results of the statistical analysis.

In particular, please consider the following

I am concerned that the results of the statistical analysis did not show significant differences and that the results of the descriptive statistics are considered mixed.

For example, the following parts of the abstract apply

Nearly all reported that the poster added new information to their knowledge (17 out of 21). The poster-exposed group showed greater improvements, comparing with the non-exposed group, on all four health outcomes, although the effects were statistically non-significant. A poster with pictorial information is effective in increasing awareness about dysphagia-specific information A poster with pictorial information is effective in increasing awareness about dysphagia-specific information among caregivers of patients in inpatient care and can be used as an augmentative means of information for an improved rehabilitation process.

I do not believe that the authors' claims of "new knowledge gained" and "significant improvement" are consistent with the lack of statistically significant differences. I presume that the reader will be confused.

From this point of view, I believe that it is better to focus the discussion specifically on the fact that no statistically significant differences were found in the discussion. Then, if future prospects are mentioned, the reader's understanding will be deepened and it will be easier to assert the significance of the authors' continued research.

That is all.

Thank you for your cooperation.

Author Response

Thank you for pointing this out. These aspects do need further clarification.

To be sure, the poster was created for the caregivers and did prove to be useful from the caregivers’ standpoint (which was the study’s main focus and primary aim), including in gained knew knowledge. The results of the descriptive statistics were not considered as mixed. The non-significant, though promising results were observed only in the additional analysis that was conducted to explore whether caregivers’ exposure to the poster also could already have some effect on patients’ dysphagia-related health. The non-significant results observed in this extra analysis were consistently presented, acknowledged, and discussed throughout the manuscript (including their possible interpretations and limitations) and also appear clearly stated (as non-significant) in the abstract. Future studies that can illuminate these results are also mentioned in the Discussion (final sentence of the second paragraph before the Conclusion).

The word “significant”, or the claim of “significant improvement”, was not used, or made, in the abstract or throughout the manuscript, in conformity with the non-significant results observed in these extra analyses that explored a possible effect of caregivers’ exposure to the poster on patients’ dysphagia-related health.

To make these issues clearer for the reader, the abstract has been changed to reinforce the primary focus of the paper (i.e., caregivers’ perspectives on the poster) and make it more clearly distinguishable from the additional analysis exploring whether caregivers’ exposure to the poster already had some influence on patients’ health parameters. Also, the word “improvement” was removed from the concluding sentence in the abstract to prevent misinterpretations. The final paragraph of the Introduction was changed accordingly.

That is all.

Thank you for your cooperation.

Thank you for your review.